# *amer1* Regulates Zebrafish Craniofacial Development by Interacting with the Wnt/β-Catenin Pathway

**DOI:** 10.3390/ijms25020734

**Published:** 2024-01-05

**Authors:** Le Sun, Lu Ping, Xinmiao Fan, Yue Fan, Bo Zhang, Xiaowei Chen

**Affiliations:** 1Department of Otolaryngology, Peking Union Medical College Hospital, Chinese Academy of Medical Sciences and Peking Union Medical College, Beijing 100730, China; pumc_sunle@student.pumc.edu.cn (L.S.); fanxinmiao@pumch.cn (X.F.); fanyue@pumch.cn (Y.F.); 2Chinese Academy of Medical Sciences and Peking Union Medical College, Beijing 100730, China; pinglu@pumch.cn; 3Department of General Surgery, Peking Union Medical College Hospital, Chinese Academy of Medical Sciences and Peking Union Medical College, Beijing 100730, China; 4Key Laboratory of Cell Proliferation and Differentiation of the Ministry of Education, College of Life Sciences, Peking University, Beijing 100871, China

**Keywords:** *amer1*, craniofacial dysplasia, cranial neural crest cells, Wnt/β-catenin pathway

## Abstract

Microtia-atresia is a rare type of congenital craniofacial malformation causing severe damage to the appearance and hearing ability of affected individuals. The genetic factors associated with microtia-atresia have not yet been determined. The *AMER1* gene has been identified as potentially pathogenic for microtia-atresia in two twin families. An *amer1* mosaic knockdown zebrafish model was constructed using CRISPR/Cas9. The phenotype and the development process of cranial neural crest cells of the knockdown zebrafish were examined. Components of the Wnt/β-catenin pathway were examined by qPCR, Western blotting, and immunofluorescence assay. IWR-1-endo, a reversible inhibitor of the Wnt/β-catenin pathway, was applied to rescue the abnormal phenotype. The present study showed that the development of mandibular cartilage in zebrafish was severely compromised by *amer1* knockdown using CRISPR/Cas9. Specifically, *amer1* knockdown was found to affect the proliferation and apoptosis of cranial neural crest cells, as well as their differentiation to chondrocytes. Mechanistically, *amer1* exerted an antagonistic effect on the Wnt/β-catenin pathway. The application of IWR-1-endo could partially rescue the abnormal phenotype. We demonstrated that *amer1* was essential for the craniofacial development of zebrafish by interacting with the Wnt/β-catenin pathway. These findings provide important insight into the role of *amer1* in zebrafish mandibular development and the pathology of microtia-atresia caused by *AMER1* gene mutations in humans.

## 1. Introduction

Microtia-atresia is a rare type of craniofacial developmental malformation characterized by shrinkage or deformity of the auricle and frequently accompanied by atresia of the external ear canal. Its incidence has been reported to range from 0.67 to 3.06 per 10,000 births [1,2]. As a manifestation of craniofacial maldevelopment, it can occur either unilaterally (70–90%) or bilaterally (10–30%), with or without middle ear malformation, and either independently (40–80%) or as a part of clinical developmental syndrome (20–60%) [3]. Etiologically, this condition is regarded as multifactorial, including both environmental and genetic factors. Although the causative genetic variations have been identified in patients with several types of syndromic microtia, the genetic variations in most patients with sporadic isolated microtia-atresia have not yet been determined.

To identify the genetic causes in patients with sporadic microtia-atresia, six families with monozygotic twins carrying discordant microtia phenotypes were subjected to whole-exome sequencing (WES) [4]. Five genes with recurrent mutations were identified, with a missense mutation (c.61C>T: p.R21C) in *AMER1*, the gene encoding APC membrane recruitment protein 1, detected in two families and predicted to be “Damaging” by both SIFT and Polyphen-2. *AMER1*, which is located on the X chromosome in humans, specifically at Xq11, has been shown to be the pathological gene responsible for osteopathia striata with cranial sclerosis (OSCS, MIM: 300373) [5], an X-linked dominant developmental disease manifesting mainly in females as macrocephaly, cleft palate, mild learning disabilities, sclerosis of the long bones and skull, and longitudinal striations visible on radiographs of the bones [6]. The disorder is usually considered lethal in males. The phenotype of microtia-atresia, however, has not been previously associated with either OSCS or *AMER1*.

Tissue-tissue interactions mediated by the Tgf-β, Wnt, Fgf, and Hh signaling molecules have been found to play key regulatory roles during the early stages of craniofacial development [7]. The Wnt signaling pathway has been shown to contribute to the development and homeostasis of craniofacial tissues [8], as well as being involved in the morphogenesis of craniofacial cartilage [9]. The protein encoded by *AMER1* was found to exert an antagonistic effect on the Wnt/β-catenin pathway [4,10]. This protein, however, is also required for the phosphorylation of the LRP6 receptor [11]. The association of the dichotomic nature of this complex interaction with craniofacial development, however, has not yet been analyzed [12,13].

Taken together, the above findings suggested that *AMER1* may play an important pathogenic role in the development of microtia-atresia. Although *amer1* has been reported to be dispensable in the development of zebrafish [14], *amer1* may be associated with early craniofacial development in the zebrafish. The present study, therefore, assessed the effects of *amer1* knockdown using CRISPR/Cas9 on abnormal phenotypes of inbred mutant offspring and on the association of these phenotypes with dysregulation of the Wnt pathway. These findings suggested that *amer1* may regulate key developmental processes and chondrogenic differentiation of cranial neural crest cells (CNCCs), and may interact with the Wnt/β-catenin pathway, showing that *amer1* may play a crucial regulatory role in early craniofacial development in zebrafish.

## 2. Results

### 2.1. Homology and Expression Profile of AMER1 in Zebrafish

Evaluation of two families having twins with microtia showed that the *AMER1* (c.61C>T: p.R21C) variant was present in both sets of twins, with both SIFT and Polyphen-2 predicting that this variant was “Damaging” and suggesting that *AMER1* may be potentially pathogenic (Figure 1A) [4]. Bioinformatics analyses also showed structural alterations in this gene and its functional association with the Wnt pathway. The protein encoded by *AMER1* was found to consist of 1135 amino acids, containing a region with six conserved block sequences and a highly divergent region. Its functional domains include those involved in phospholipid binding, APC interaction, and β-catenin binding. Evaluation of genes homologous to human *AMER1* in zebrafish by NCBI BLAST (https://blast.ncbi.nlm.nih.gov/Blast.cgi (accessed on 3 December 2022)) (Figure 1B) identified a zebrafish gene encoding a 953-amino acid protein. The human and zebrafish protein sequences were 35.91% identical, with an expected value of 2 × 10^−70^, indicating a high degree of homology.

The spatiotemporal pattern of *amer1* expression during zebrafish embryonic development was examined using two approaches. The expression of *amer1* during the early stages of zebrafish embryonic development was evaluated initially using the recently established ZESTA database (https://db.cngb.org/stomics/zesta/ (accessed on 19 December 2022)) (Figure 1C, Appendix A). Examination of the scRNA data showed that, at 10 h post-fertilization (hpf), *amer1* was expressed in the anterior neural keel and neural plate. At 12 hpf, *amer1* was mainly expressed in the neural crest and neural rod, whereas, at 18 and 24 hpf, *amer1* was expressed in the otic placode, otic cup, and otic vesical at 18 and 24 hpf (Figure 1C). Evaluation of the stereoseq data at six time points (3, 5.25, 10, 12, 18, 24 hpf) yielded results similar to those of the scRNA data, with *amer1* mainly expressed in the otic vesicle at 18 and 24 hpf (Appendix A).

To evaluate the in vivo expression profile of *amer1* in zebrafish, RNA was extracted from zebrafish embryos at different developmental stages and cDNA libraries were constructed. The expression of *amer1* could be detected beginning at the one-cell stage up to 7 days post-fertilization (dpf) (Figure 2A). To confirm these findings and to broaden the time window for expression, embryos during the first 6 days of development were subjected to whole-mount in situ hybridization (Figure 2B–K). Expression of *amer1* was detected in zebrafish ova, indicating maternal expression of this gene, similar to findings in the ZESTA database (Figure 2B). At 18 hpf, *amer1* expression could be observed in the otic vesicle (Figure 2F). At 1 dpf, *amer1* was detected in the pharyngeal arches (Figure 2G), with this gene being mainly expressed in the pharyngeal arches beginning at 48 hpf (Figure 2H–K), indicating that this gene plays a role in pharyngeal development.

### 2.2. Knockdown of amer1 Results in Severe Pharyngeal Malformation

To assess the effects of *amer1* variants on pharyngeal development, *amer1* expression was knocked down in zebrafish using the CRISPR/Cas9 technique [15]. Four gRNAs were designed and validated (Figure 3A–E, Appendix A), with each co-injected with Cas9 into one-cell embryos. Evaluation of representative *amer1* knockdown embryos at 1 dpf showed that the four gRNAs had efficacies of 67.0%, 83.6%, 96.3%, and 89.2%, respectively.

The gRNA (R3) with the highest knockdown efficacy was selected for the construction of a zebrafish *amer1* knockdown model. The phenotypes of the wildtype and *amer1* mosaic knockdown zebrafish were compared (Figure 3F–M). The *amer1* knockdown embryos were observed to have no obvious phenotype anomalies during the early developmental stages (Figure 3F,J). At 3 dpf, however, when the wildtype embryos began to form the structure of the lower jaw, the *amer1* knockdown embryos abnormally lacked visible signs of mandible development (Figure 3G,K). These differences were even more apparent after 4 dpf, as the bladder began to inflate and the lower jaw continued to protrude forward in wildtype embryos, whereas neither the bladder nor the lower jaw showed signs of development in the *amer1* knockdown embryos (Figure 3H,L). The differences persisted until 6 dpf, with all abnormal zebrafish dying of breathing and/or swallowing difficulties (Figure 3I,M). The otocysts and otoliths of wildtype and *amer1* knockdown embryos were essentially the same, however, indicating that the differences in lower jaw and bladder development were not caused by overall developmental delays, but by defects in the development of these specific tissues.

Differences in mandible development between wildtype and *amer1* knockdown embryos were further demonstrated by Alcian staining of pharyngeal arch cartilage from complete and sliced embryos at 4 dpf (Figure 4A–P). Partial loss and malformation of Meckel’s and the palatoquadrate cartilage were observed in the *amer1* knockdown embryos (Figure 4A–D). Staining of the slices more clearly demonstrated the deformed cartilages (Figure 4M–O). To further examine the structure of pharyngeal chondrocytes at a cellular level, cartilage samples were also stained with wheat germ agglutinin (WGA) (Figure 4Q–U). Chondrocytes in Meckel’s cartilage of *amer1* knockdown embryos were swollen and their alignment was disoriented (Figure 4S–U), indicating severe defects in cartilage development. To further validate that loss of *amer1* is the cause of the abnormal pharyngeal arch development, we performed a rescue experiment with *amer1* mRNA and observed a complete reversal of the abnormal phenotype (Appendix A).

### 2.3. Inbred Homozygous Zebrafish Demonstrates Similar Phenotype after Excluding Potential Compensation

To determine if the abnormal pharyngeal trait could be passed on, F0 mosaic knockdown zebrafishes were mated to obtain stable heterozygotic *amer1*^+/−^ F1 fishes. These fishes showed no apparent developmental anomalies, and their genotypes were validated. F1 males and females with the same genotype were mated to obtain *amer1*^−/−^ F2 fishes, with genotyping showing that approximately 1/4 of these F2 fishes were homozygotic *amer1*^−/−^. However, these *amer1*^−/−^ fishes did not demonstrate visible malformation during early developmental stages (0 hpf to 6 dpf) (Figure 5A). WGA staining of their cartilage also showed no apparent deformation or disorderly arrangement of their chondrocytes (Figure 5B). Examination of the altered DNA sequence “TCGTGA” showed that an extra C was present in the encoded mRNA, causing a shift of the reading frame and a pre-mature “UGA” termination codon. Because a compensation effect may have been triggered [16], *Upf3a* morpholino was injected into *amer1*^−/−^ embryos to mitigate potential genetic compensatory effects. Sole injection of *upf3a* morpholino into wildtype embryos did not induce apparent anomalies. After injection, the offspring demonstrated a 3:1 phenotypic segregation of lower jaw deformity vs. normal fishes (Figure 5A). Fluorescence imaging revealed swallowed and misaligned chondrocytes in *amer1*^−/−^ embryos treated with *upf3a* morpholino (Figure 5B). Unfortunately, *amer1*^−/−^ zebrafish could not stably pass on this genotype, as these fish died of breathing and swallowing difficulties after 6 dpf.

### 2.4. Knockdown of amer1 Affects Chondrocyte Differentiation, Proliferation, and Apoptosis

Because malformation of multiple cartilages was observed in *amer1* knockdown embryos, the process by which chondrocytes develop from CNCCs was evaluated in zebrafish embryos. Determination of the numbers of CNCCs in wildtype and F0 *amer1* knockdown embryos at 24 hpf showed that the proliferation of CNCCs, as determined by staining for PH3, was significantly lower in *amer1* knockdown than in wildtype embryos (17.00 ± 3.46 vs. 69.67 ± 8.62 cells per fish, *p* = 0.0006) (Figure 6). Conversely, the number of cells that had undergone apoptosis, as determined by TUNEL staining, was significantly higher in *amer1* knockdown than in wildtype embryos (197.33 ± 35.36 vs. 28.33 ± 7.50 cells per fish, *p* = 0.0013) (Figure 6).

To evaluate the effects of *amer1* knockdown on the specific developmental stages from CNCCs to chondrocytes, marker genes for CNCC development were assayed in zebrafish embryos of various developmental stages by in situ hybridization. The levels of expression of *crestin* and *foxd3* were similar in wildtype and F0 *amer1* knockdown embryos at 12 hpf, indicating that CNCC formation was not affected by *amer1* knockdown (Figure 7A,B). The levels of expression of *dlx2a* in wildtype and F0 *amer1* knockdown embryos at 30 hpf also showed no apparent differences, suggesting that *amer1* was not necessary for the specialization of CNCCs to pharyngeal arch CNCCs (Figure 7C). The area of expression of *barx1* was smaller in *amer1* knockdown than in wildtype embryos at 48 hpf, although their expression patterns were similar, indicating that *amer1* knockdown had a slight effect on the number of CNCCs rather than on the coagulation of mesenchymal cells (Figure 7D). Expression of the chondrogenic markers *sox9a* and *col2a1a* in wildtype and F0 *amer1* knockdown embryos at 72 hpf differed significantly, with *sox9* being significantly upregulated (*p* = 0.0007 for lateral view and *p* = 0.0441 for dorsal view) and *col2a1a* significantly downregulated (*p* = 0.0092 for lateral view and *p* = 0.6436 for dorsal view) in F0 *amer1* knockdown embryos (Figure 7E–H). Taken together, these results showed that chondrocyte differentiation and collagen formation were partially blocked by *amer1* knockdown.

### 2.5. Amer1 Exerts Its Effects via the Wnt/β-Catenin Signaling Pathway

Because the *amer1* gene has been associated with the Wnt/β-catenin pathway, a vital regulatory pathway in early craniofacial and pharyngeal arch development, *amer1* knockdown may cause malformations in *amer* knockdown embryos by interfering with the wnt/β-catenin pathway. Western blotting of full-embryo proteins extracted from wildtype and F0 *amer1* knockdown embryos at 4 dpf showed that β-catenin expression was significantly higher in *amer1* knockdown embryos (Figure 8A,B). Evaluation by qPCR of genes acting downstream of β-catenin showed that the levels of expression of several genes, including *lef1*, *jun*, and *fosl1a*, were significantly higher (*p* = 0.0069 for *lef1*, *p* = 0.0020 for *jun*, *p* = 0.0113 for *fosl1a*) in *amer1* knockdown embryos (Figure 8C). Moreover, in situ hybridization showed that *lef1* and *jun* expression was upregulated in the mandible region of *amer1* knockdown embryos at 2 and 3 dpf (Figure 8D–G).

Upregulation of these canonical TCF/β-catenin target genes suggested the possible translocation of β-catenin. Immunofluorescence staining of tissue sections from wildtype and *amer1* knockdown embryo sections of the ceratohyal cartilage plane at 4 dpf showed that β-catenin was ubiquitously expressed in the cytoplasm of cells of the ceratohyal cartilage of wildtype embryos, but co-localized with the nucleus of mutant embryos (Figure 8H,I).

### 2.6. An Inhibitor of the Wnt/β-Catenin Signaling Pathway Rescues the Malformation of amer1 Knockdown Embryos

To further explore the causative role of dysregulation of the Wnt/β-catenin signaling pathway in the pathogenicity of *amer1* mutation, F0 *amer1* knockdown embryos were treated with IWR-1-endo, a reversible inhibitor that promotes β-catenin phosphorylation by interacting with axin, and observed whether IWR-1-endo could rescue embryos from the malformed phenotype. Mutant embryos at 12, 24, and 48 hpf were treated with 5 and 10 μM IWR-1-endo for 24 h; the inhibitor was removed; and the embryos were fixed and observed at 96 hpf (Figure 9A). Treatment of embryos at 12 and 24 hpf with 5 μM IWR-1-endo for 24 h resulted in phenotypic salvage of more than 80% of embryos, with application starting at 24 hpf being the most effective dose with the lowest toxicity. The malformation rate of *amer1* knockdown embryos was significantly lower in the presence that in the absence of IWR-1-endo (20.50% vs. 40.50%, *p* = 0.009), indicating successful rescue (Figure 9B), with similar results observed on representative brightfield images (Figure 9C). Alcian staining of the pharyngeal arch cartilage in 100 *amer1* knockdown embryos treated with IWR-1-endo at 4 dpf showed that cartilage development could be divided into five categories: normal (*n* = 53), mild dysplasia (*n* = 21), partial loss (*n* = 15), complete loss (*n* = 3), and others (including dead embryos) (*n* = 8) (Figure 9D). Western blotting of β-catenin and qPCR of canonical TCF/β-catenin target genes demonstrated that the expression levels of β-catenin, lef1, jun, and fosl1a were all reduced by IWR-1-endo treatment, to concentrations similar to those in wildtype embryos (Figure 10A–C). Immunofluorescence staining demonstrated that co-localization of β-catenin with the nucleus in *amer1* knockdown embryos was lower in the presence than in the absence of IWR-1-endo (Figure 10D,E).

## 3. Discussion

Because both environmental and genetic factors may contribute to the etiology of microtia, it is difficult to investigate these factors separately, especially in patients with sporadic microtia and no apparent family history. This issue was previously addressed by recruiting six families with monozygotic twins having discordant microtia/normal phenotypes [4]. These twins grew in the same environment and their genotypes were nearly identical, enabling isolation of a genetic component contributing to the etiology of microtia. The findings of that study suggested that *AMER1* (c.61C>T: p.R21C) may be a pathogenic genetic variant of *AMER1*, as not only was this variant observed in more than one sporadic family, but it resulted in both structural alteration and predicted functional damage.

Previous studies have assessed the consequences of the loss-of-function of *AMER1* in zebrafish and mice, but these results were inconsistent [14,17,18]. Knockdown of *Amer1* in mice resulted in neonatal lethality, somatic overgrowth, and deficiencies in multiple mesenchyme-derived tissues [17], and a previous study of morpholino-based *amer1* knockdown in zebrafish revealed abnormal phenotypes, including anterior truncations, epiboly defects, headless malformations, and small eyes [18]. In contrast, another study in zebrafish based on morpholino knockdown and TALEN knockdown techniques found no apparent abnormal phenotypes, suggesting that *amer1* was dispensable in zebrafish [14]. The present study used the CRISPR/Cas9 strategy and four gRNAs to knock down *amer1* expression, with the gRNA having the highest efficiency being chosen for further evaluation. Morpholino-based knockdown of *amer1* and inbred fish lines were both utilized to validate phenotypic findings, with the abnormal phenotype in zebrafish being homologous to the malformations observed in patients with microtia [4]. These findings suggest that *amer1* can regulate embryonic and craniofacial development in zebrafish.

Three major hypotheses to date have been associated with microtia-atresia and its related craniofacial malformations: vascular abnormality, damage to Meckel’s cartilage, and disruption of CNCC development [19]. *Amer1* knockdown in mice has been reported to affect multiple mesenchyme-derived tissues, suggesting that this gene affects the development of CNCCs, which form a mesenchymal core between the ectoderm and endoderm along with mesoderm cells and play an important role in the development of cartilage [17]. PH3 and TUNEL staining differed significantly between *amer1* knockdown and control zebrafish, suggesting that *amer1* knockdown reduced the numbers of CNCCs by reducing cell proliferation and enhancing apoptosis. In situ hybridization assessing the expression of marker genes during each step of CNCC development showed that the expression of *sox9a* was significantly upregulated and that of *col2a1a* was downregulated in *amer1* knockdown zebrafish. During chondrogenesis, *sox9a* and *col2a1* are normally expressed in parallel, with *Col2a1* acting directly downstream of *sox9a* [20,21]. The findings of the present study suggest that *amer1* knockdown affected the differentiation of CNCCs to chondrocytes, with the upregulation of *sox9a* likely caused by a negative feedback mechanism.

Studies have evaluated the relationship between *AMER1* and the Wnt/β-catenin pathway. For example, *AMER1* was found to inhibit the Wnt pathway through its assembly of the β-catenin destruction complex at the plasma membrane, resulting in the induction of β-catenin degradation, with in vitro knockdown of *AMER1* in cell lines resulting in the activation of Wnt target genes [22]. AMER1, however, has also been shown to act as a scaffold protein to stimulate LRP6 phosphorylation, a key step in Wnt/β-catenin signaling, with knockdown of *AMER1* reducing Wnt-induced LRP6 phosphorylation [23]. These dichotomic regulatory roles of *AMER1* provide an answer to its enigmatic relationship with the Wnt/β-catenin pathway [11].

Although interactions between *AMER1* and the Wnt/β-catenin pathway have been explored during the process of tumorigenesis, less is known about this relationship during embryonic and craniofacial development. Defective cell fate determination caused by *Amer1* deficiency in mice was shown to be due to aberrant β-catenin activation [17], and silencing of *amer1* in zebrafish was found to activate Wnt/β-catenin reporter genes [18]. The current study assessed whether malformations observed in zebrafish had similar underlying mechanisms. Many of the genes in the Wnt/β-catenin pathway, including *lef1*, *jun*, and *fosl1a*, were found to be significantly upregulated in *amer1* mutant zebrafish. A previous study in mice concerning Lmp1 found that Amer1 was downregulated, accompanied by the cytoplasmic enrichment and endonuclear accumulation of β-catenin [13]. This finding was in agreement with the results of the present study, showing that decreased *amer1* expression in zebrafish hindered the destruction of β-catenin, leading to its accumulation and translocation from the cytoplasm to the cell nucleus. Taken together, these findings support the hypothesis that *amer1* negatively regulates the Wnt/β-catenin pathway during the early stages of embryonic development and *amer1* mutation aberrantly activates this pathway, resulting in craniofacial malformation.

To further show that the abnormal phenotype induced by *amer1* mutation was due to abnormal activation of the Wnt/β-catenin pathway, two types of rescue analyses, both targeting this pathway, were performed. Treatment of zebrafish embryos with IWR-1-endo inhibitor or expression of its mRNA downregulated the Wnt/β-catenin pathway to near normal levels, as well as reducing the percentage of abnormal embryos. In addition to showing that *amer1* regulates the Wnt/β-catenin pathway, these findings support the significance of this pathway in the development of microtia and suggest potential therapeutic targets.

## 4. Materials and Methods

### 4.1. Zebrafish and Embryos

Adult zebrafish were maintained under standard conditions [24], and embryos were morphologically staged using standard methods [25]. Tuebingen and the transgenic fish line, *Tg* (*sox10:EGFP*) (ID:CZ156, ba2Tg/C) [26], were acquired from China Zebrafish Resource Center (CZRC) (http://www.zfish.cn/ (accessed on 1 May 2021)). All zebrafish experiments were approved by the Animal Ethics Committee of Peking University (protocol code: LSC-ZhangB-3, date of approval: 1 September 2019) and conformed to institutional, local, and national rules and guidelines. All phenotypic experiments were carried out within 5 days post fertilization (dpf).

### 4.2. Photography and Image Processing

A Stemi 2000-C optical stereomicroscope (Zeiss, Oberkochen, Germany) was used for all brightfield observations, including in situ hybridization. Brightfield images were taken with a Zeiss AxioCam MRc5 camera. Immunofluorescence images were taken with a Zeiss LSM 710 NLO laser scanning confocal microscope and Duoscan System. Images were processed and analyzed by Zeiss Zen 2009.

### 4.3. Bioinformatic Prediction of amer1 Expression Profile in Zebrafish

The expression of *amer1* during the early stages of zebrafish embryonic development was evaluated in the Spatial Transcript Omics DataBase (STOmics DB) (https://db.cngb.org/stomics/ (accessed on 19 December 2022)) [27], both in single-cell RNA (scRNA) sequencing and stereo-sequencing (stereoseq) data. Bin annotation was applied for analysis of stereoseq data.

### 4.4. Whole-Mount In Situ Hybridization

RNA probes were transcribed from linearized plasmids using the proper RNA polymerase (Roche Diagnostics, Tokyo, Japan; Stratagene Japan, Tokyo, Japan). Whole-mount in situ hybridization and immunostaining were performed as described in [28]. Embryos were fixed overnight in 4% paraformaldehyde, followed by dehydration in methanol and storage in methanol at −20 °C. Primers for in situ hybridization included those for *amer1* (forward, 5′-GTACCCAACAGTGACGAGGG-3′; reverse, 5′-GGGCCAAAGGGAGAACTTGA-3′), and for *crestin*, *foxd3*, *dlx2a*, *barx1*, *sox9*, *col2a1a*, *fgf3*, *tbx1*, and *nkx2.3* (Appendix A).

### 4.5. Genetic Manipulation of amer1

Knockdown of *amer1* was performed using the CRISPR/Cas9 system [15]. Four different sets of guide RNAs (gRNAs, Appendix A) were designed and co-injected with Cas9 into 1-cell embryos. The gRNAs were designed prior to the construction of the Genome-Scale Guide Set using the CHOPCHOP website (https://chopchop.rc.fas.harvard.edu (accessed on 11 June 2021)) [29]. gRNAs were generated by PCR with a forward primer containing a T7 promoter, the guide sequence, a scaffold sequence, and a reverse primer encoding the standard chimeric gRNA scaffold “tracer rev” [30]. The efficacy of gRNA was verified by extracting crude genomic DNA from whole zebrafish embryos, followed by PCR amplification with the indicated primers (Appendix A) and sequencing. A morpholino targeting *upf3a* was also injected at the dosage of 4 pg per embryo to inhibit the compensatory effects of DNA damage [31]. An *amer1*-targeting morpholino (5′-CGATCTCCATAATGACACAGGTGAC-3′), synthetized by Gene Tools, LLC, was applied to knockdown *amer1* aiming to replicate and to confirm the phenotypic changes associated with *amer1* knockdown. Homozygotic embryos were bred, whereas unless stated otherwise, F0 *amer1* knockdown embryos were used in most of the experiments.

### 4.6. Cartilage Staining and Immunofluorescence Staining

Embryos were fixed in 4% paraformaldehyde (PFA) at 4 °C overnight, washed in phosphate-buffered saline containing 0.1% Tween-20 (PBST) for 8 h, and subjected to Alcian blue staining, as described in [32]. Embryos were also fixed and stained with wheat germ agglutinin (WGA; Invitrogen, Carlsbad, CA, USA) and 4′,6-diamidino-2-phenylindole (DAPI; Sigma-Aldrich, St. Louis, MO, USA) [33].

*Tg (sox10:EGFP)* zebrafish embryos were fixed with 4% phosphate-buffered paraformaldehyde, washed with PBST for 20 min, and immunostained with antibody to phosphohistone H (PHH3; 1:400; sc-374669, Santa Cruz Biotechnology, Santa Cruz, CA, USA). Terminal deoxynucleotidyl transferase dUTP nick-end labeling (TUNEL) assays were performed using the In Situ Cell Death Detection Kit, TMR red (12156792910, Roche, Basel, Basel state, Switzerland), according to the manufacturer’s instructions. Nuclei were visualized by staining with DAPI; proliferation by staining with anti-PHH3; and apoptosis by TUNEL assays. To ensure that only cranial neural crest cells (CNCCs) were assessed, only cells that were doubly positive for anti-PHH3/TUNEL and EGFP were counted manually.

In the immunofluorescence experiments, embryos were sectioned at the plane of ceratohyal cartilage. Cells were stained with monoclonal mouse anti-β-catenin antibody (1:500, C7207, Sigma-Aldrich, St. Louis, MO, USA), followed by staining with Alexa Fluor 647-conjugated goat anti-mouse IgG H&L (1:500, ab150115, Abcam, Cambridge, UK) or Alexa Fluor 594 conjugated WGA (5 μg/mL, W11262, ThermoFisher, Waltham, MA, USA).

### 4.7. Statistical Analysis

In all phenotypic observational experiments, at least 20 embryos were observed in each condition. Our experiments showed congruent results, affecting >85% of embryos observed, and all selected images are representative. All statistical analyses were carried out using 3 randomly picked embryos in each group and statistical significance was assessed by unpaired *t*-tests using GraphPad Prism 6.0 (GraphPad Software, San Diego, CA, USA), with *p* < 0.05 considered statistically significant.

## 5. Conclusions

In conclusion, the present study identified a gene in zebrafish that was homologous to human *AMER1*, a candidate gene associated with the pathogenesis of microtia-atresia. The expression profile of *amer1* in zebrafish embryos was determined by RT-PCR and in situ hybridization. Genetic manipulation of *amer1* in zebrafish using the CRISPR/Cas9 technique resulted in severe pharyngeal malformation and cartilage deformity, with a stable inbred fish line established after excluding potential compensation effects. Chondrocyte differentiation, proliferation, and apoptosis were all affected by *amer1* variation. Variations in *amer1* upregulated the expression of multiple proteins of the Wnt/β-catenin signaling pathway, as well as altering the intracellular distribution of β-catenin. Treatment of zebrafish embryos with an inhibitor of the Wnt/β-catenin signaling pathway successfully rescued the malformation of the mutant embryos. Taken together, these results support the hypothesis that *amer1* regulates zebrafish craniofacial development by interacting with the Wnt/β-catenin pathway.

## Figures and Tables

**Figure 1 ijms-25-00734-f001:**
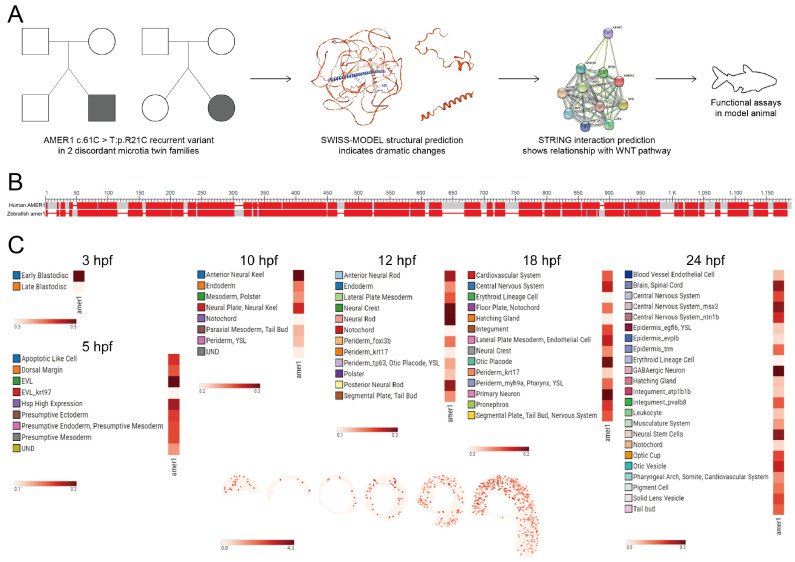
Identification of *AMER1* and its homologous gene in zebrafish. (**A**) Schematic diagram of *AMER1* sequences in families of patients with dichotomous monozygotic twins with microtia. The *AMER1* variant (c.61C>T: p.R21C) was detected in two patients with sporadic microtia (shaded), as well as in their normal twins and parents (blank). The predicted protein structure showed dramatic changes from wildtype, and pathway prediction showed a relationship between *AMER1* and the Wnt pathway. (**B**) BLAST analysis of sequences homologous to human *AMER1* in zebrafish. (**C**) Expression profile of zebrafish *amer1*, as determined using the Spatial Transcript Omics DataBase.

**Figure 2 ijms-25-00734-f002:**
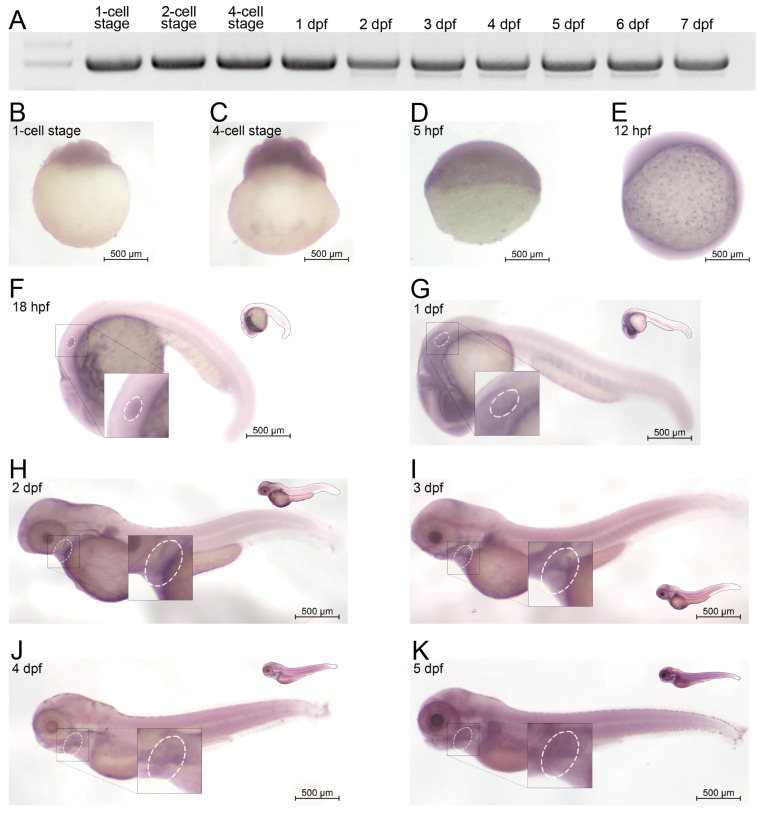
Expression of *amer1* in zebrafish. (**A**) RT-PCR analysis, showing that *amer1* was constitutively expressed in zebrafish embryos, beginning at the 1-cell stage to 7 dpf. (**B**–**K**) In situ hybridization of *amer1*, showing that (**F**) *amer1* was expressed in otic vesicles (white circle) at 18 hpf; and that (**H**–**K**) concentrated expression of *amer1* in the mandible region (white circle) was observed beginning at 2 dpf.

**Figure 3 ijms-25-00734-f003:**
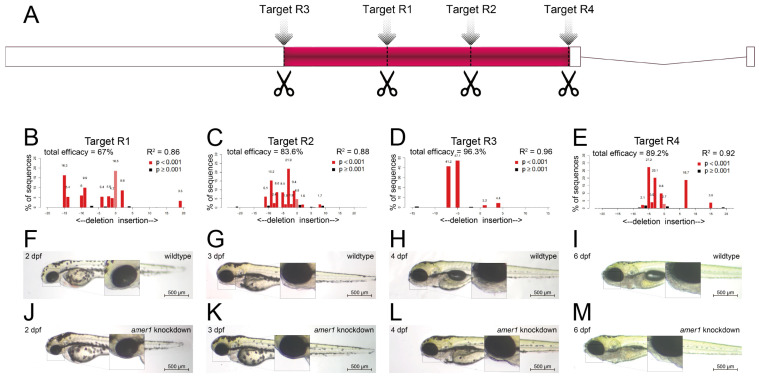
Knockdown of *amer1* and phenotypic changes. (**A**) Schematic diagram showing the four *amer1* gRNA targets of the CRISPR/Cas9 knockdown system. (**B**–**E**) PCR analysis, showing representative *amer1* knockdown efficiencies of the four gRNA targets. (F-M) Brightfield images of embryos of (**F**–**I**) wildtype and (**J**–**M**) *amer1* knockdown zebrafish at (**F**,**J**) 2, (**G**,**K**) 3, (**H**,**L**) 4, and (**I**,**M**) 6 dpf, showing malformed mandibles in the *amer1* knockdown embryos.

**Figure 4 ijms-25-00734-f004:**
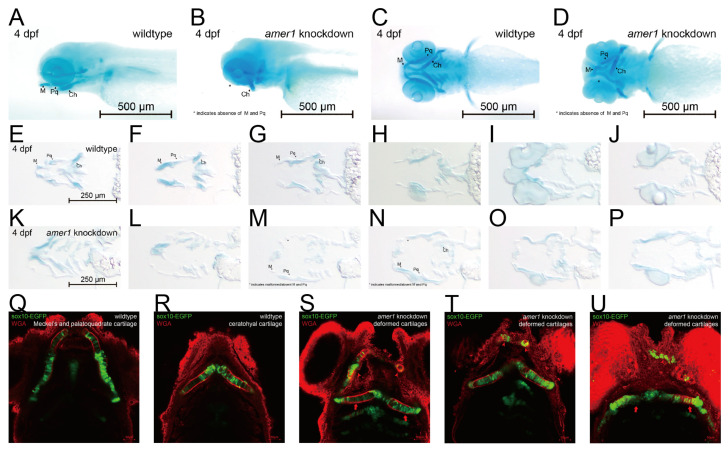
Phenotypic changes induced by *amer1* knockdown, as shown by Alcian and WGA staining. (**A**–**D**) Alcian staining of embryos of (**A**,**C**) wildtype and (**B**,**D**) *amer1* knockdown embryos at 4 dpf, showing malformation of Meckel’s and palatoquadrate cartilages in the *amer1* knockdown embryos. (**E**–**P**) Alcian staining of tissue sections of embryos of (**E**–**J**) wildtype and (**K**–**P**) *amer1* knockdown zebrafish at 4 dpf, showing that Meckel’s cartilage was severely deformed in *amer1* knockdown embryos (red rectangles in (**M**–**O**)). (**Q**,**R**) WGA staining of wildtype embryos, showing regularly distributed, well-shaped chondrocytes, with Meckel’s cartilage and ceratohyal cartilage normally not observed in the same plane. (**S**–**U**) WGA staining of *amer1* knockdown embryos, showing Meckel’s cartilage (red arrowheads) and palatoquadrate and ceratohyal cartilage (long red arrows) in the same plane. All cartilage was deformed and the arrangement of chondrocytes was disordered.

**Figure 5 ijms-25-00734-f005:**
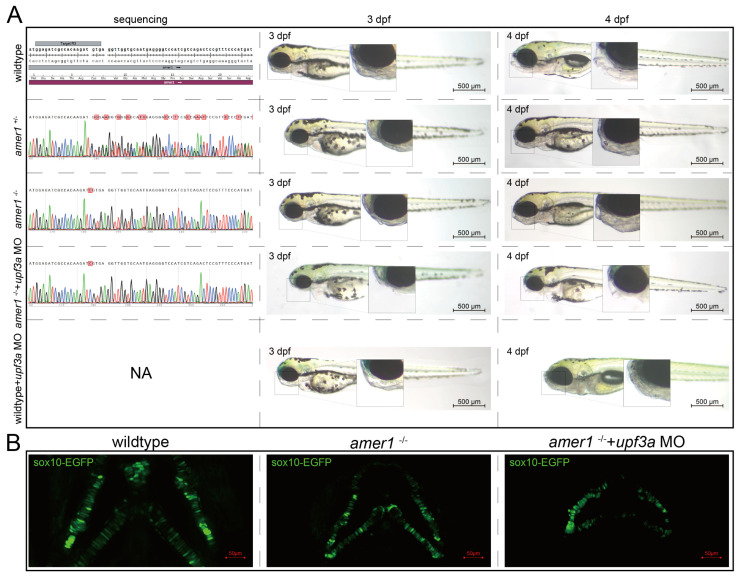
Genotypes and phenotypes of F2 zebrafish. (**A**) Sequencing results and brightfield images of F2 zebrafishes at 3 and 4 dpf with five different genotypes/treatments: wildtype, *amer1*^+/−^, *amer1*^−/−^, and *amer1*^−/−^ + *upf3a* MO, wildtype + *upf3a* MO. *amer1*^−/−^ F2 embryos treated with *upf3a* MO showed malformation of the mandible. Red suqares indicate mutated sites. NA: not applicable. (**B**) Fluorescence staining showing that chondrocytes were regularly distributed in wildtype and *amer1*^−/−^ F2 embryos, but were disarranged in *amer1*^−/−^ F2 embryos treated with *upf3a* MO.

**Figure 6 ijms-25-00734-f006:**
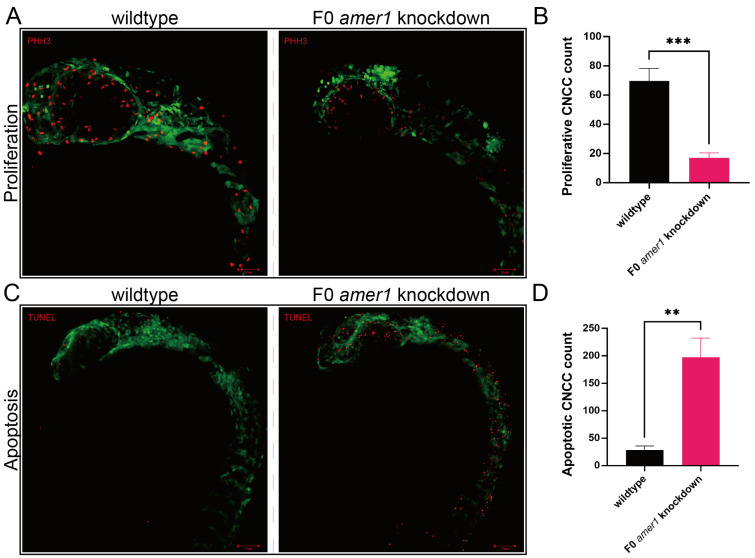
Proliferation and apoptosis of CNCCs in zebrafish embryos. (**A**) Proliferation of CNCCs, as shown by PHH3 staining, in wildtype and F0 *amer1* knockdown embryos. (**B**) Quantitation of CNCC proliferation in wildtype and *amer1* knockdown embryos (*n* = 3 each). (**C**) Apoptosis of CNCCs, as shown by TUNEL staining. (**D**) Quantitation of CNCC apoptosis in wildtype and *amer1* knockdown embryos (*n* = 3 each). ** *p* < 0.01; *** *p* < 0.001.

**Figure 7 ijms-25-00734-f007:**
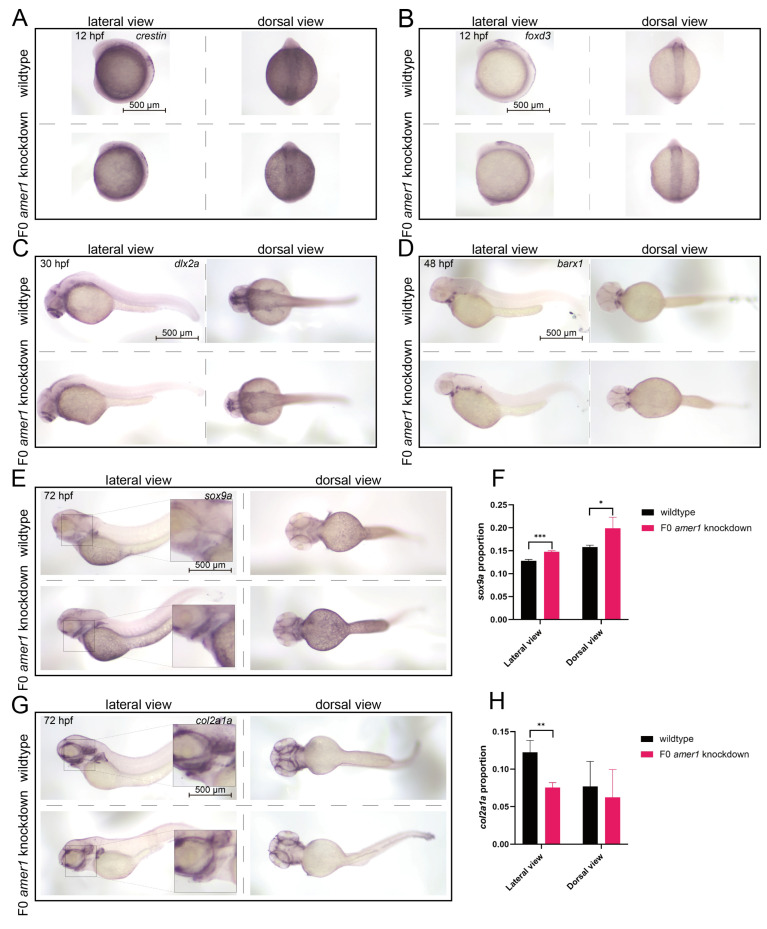
Expression of CNCC related markers in zebrafish embryos. In situ hybridization, showing the expression of (**A**) *crestin* and (**B**) *foxd3* at 12 hpf, (**C**) *dlx2a* at 30 hpf, and (**D**) *barx1* at 48 hpf in wildtype and F0 *amer1* knockdown embryos. (**A**–**C**) The levels of expression of *crestin*, *foxd3*, and *dlx2a* showed no apparent differences in wildtype and *amer1* knockdown embryos. (**D**) The expression pattern of *barx1* was smaller in *amer1* knockdown than in wildtype embryos. (**E**,**F**) The expression of *sox9a* at 72 hpf was increased in *amer1* knockdown than in wildtype embryos (for quantitative analysis, *n* = 3 each). (**G**,**H**) The expression of *col2a1a* at 72 hpf was decreased in *amer1* knockdown than in wildtype embryos (for quantitative analysis, *n* = 3 each). * *p* < 0.05; ** *p* < 0.01; *** *p* < 0.001.

**Figure 8 ijms-25-00734-f008:**
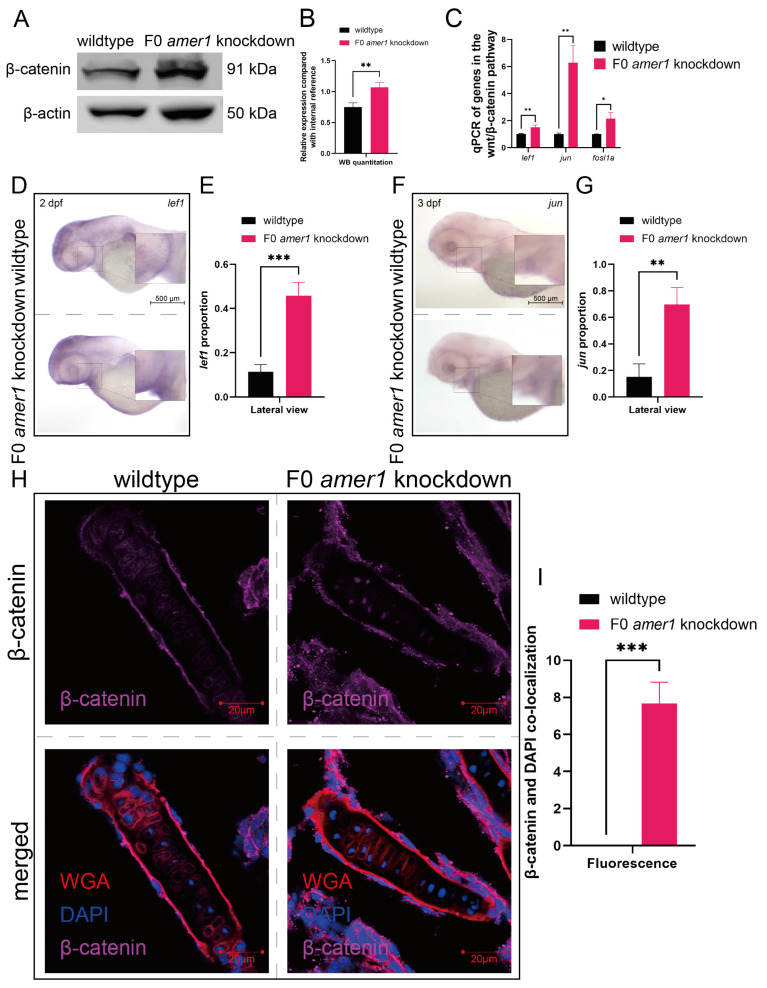
Expression and localization of components of the wnt pathway in zebrafish embryos. (**A**,**B**) Western blotting and quantitation showing that β-catenin levels were higher in *amer1* knockdown than in wildtype embryos at 4 dpf. (**C**) qPCR showing that the levels of expression of *lef1*, *jun*, and *fosl1a* were significantly higher in *amer1* knockdown embryos. (**D**–**G**) Validation of the expression of (**D**) *lef1* and (**F**) *jun* by in situ hybridization and their quantitation (**E**,**G**) (for quantitative analysis, *n* = 3 each). (**H**) Immunofluorescence staining of the ceratohyal cartilage using WGA, β-catenin, and DAPI in wildtype and *amer1* knockdown embryo sections at 4 dpf. (**I**) Manual counting of co-localization of β-catenin with cell nuclei (*n* = 3 each). * *p* < 0.05; ** *p* < 0.01; *** *p* < 0.001.

**Figure 9 ijms-25-00734-f009:**
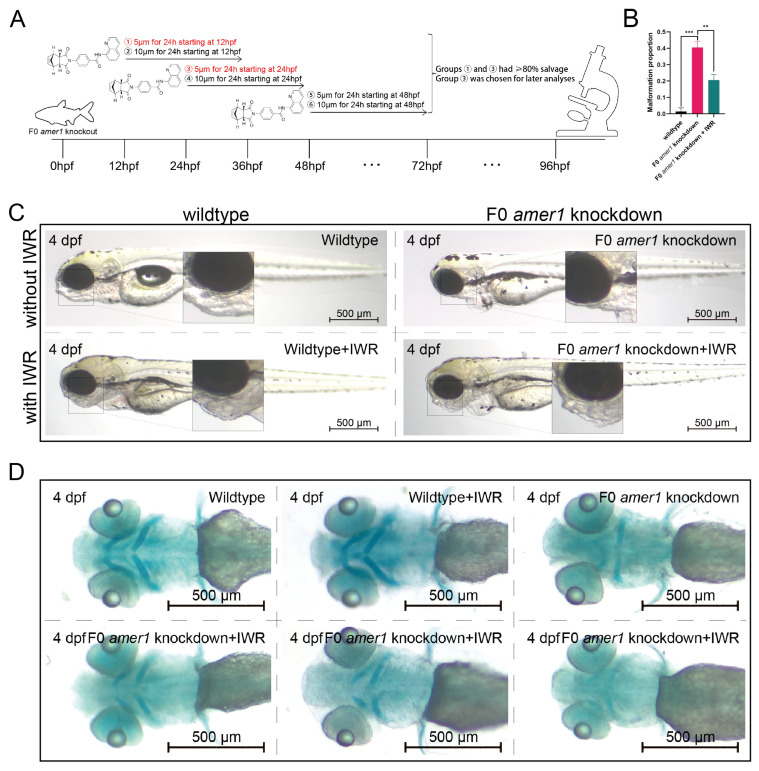
Phenotypic rescue of F0 *amer1* knockdown embryos using IWR-1-endo. (**A**) Schematic diagram of the rescue and observation process. Different numbers indicate different treatment. (**B**) Proportions of malformed wildtype, *amer1* knockdown and IWR-treated *amer1* knockdown embryos, showing that the proportion of malformed IWR-treated *amer1* knockdown was significantly lower than that of *amer1* knockdown embryos. (**C**) Representative brightfield images of wildtype, *amer1* knockdown embryos, and IWR-treated *amer1* knockdown embryos. (**D**) Representative Alcian staining images of wildtype (**first row**, **left**), IWR-treated wildtype (**first row**, **middle**), *amer1* knockdown (**first row**, **right**), and IWR-treated *amer1* knockdown embryos showing normal development of pharyngeal arch cartilage (**second row**, **left**), partial loss (**second row**, **middle**), and complete loss (**second row**, **right**). ** *p* < 0.01; *** *p* < 0.001.

**Figure 10 ijms-25-00734-f010:**
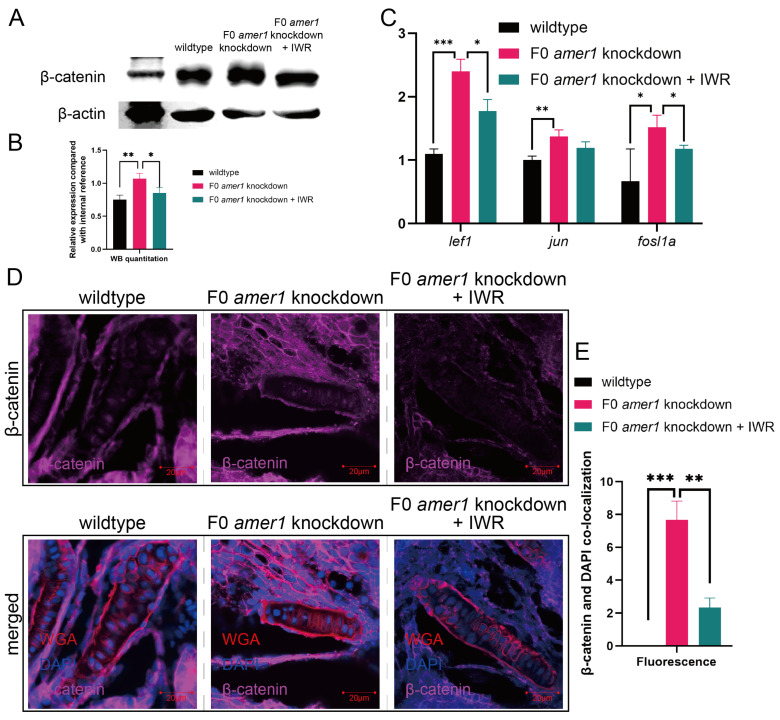
Effects of IWR-1-endo on components of the Wnt pathway in F0 *amer1* knockdown embryos. (**A**,**B**) Western blotting and quantitation showing that β-catenin expression was lower in IWR-treated *amer1* knockdown embryos than in *amer1* knockdown embryos at 4 dpf. (**C**) qPCR showing that the levels of expression of *lef1*, *jun*, and *fosl1a* were lower in IWR-treated than in untreated *amer1* knockdown embryos. (**D**) Co-localization of β-catenin with the cell nucleus in the ceratohyal cartilage was reduced in *amer1* knockdown embryos treated with IWR. (**E**) Manual counting of co-localization of β-catenin with cell nuclei in embryos with different treatments (*n* = 3 each). * *p* < 0.05; ** *p* < 0.01; *** *p* < 0.001.

## Data Availability

All figures are associated with raw data. Raw images can be provided upon request. Sequences used for CRISPR/Cas9 and for in situ hybridization are included in the Appendix A.

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
