# Peer review of "amer1 Regulates Zebrafish Craniofacial Development by Interacting with the Wnt/β-Catenin Pathway"

_ijms, 2024, doi:10.3390/ijms25020734_

Round 1

Reviewer 1 Report

Comments and Suggestions for Authors

In the article entitled “amer1 Regulates Zebrafish Craniofacial Development by Inter-2 acting with the Wnt/β-catenin Pathway” the authors investigate the function of amer1 with early craniofacial development in zebrafish. They postulate that amer1 is essential for craniofacial development in zebrafish and interacts with Wnt signaling. While this work is significant, it lacks scientific rigor, and several major points need to be addressed prior to publication:

·      Line 158: while the authors describe a heterozygous animal (amer1+/-), they state several times throughout the paper, amer1+/- fish do not have a phenotype. In addition, this description does not match legend of figure 4.

·      Labels for most figures are too small and hard to read (Figure 2, 3, 4, 5, 7,8)

·      The authors need to include in Figure 5, wild-type zebrafish injected with upf3a MO as changes could be caused by injection of this MO.

·      Figure 6 to Figure 10: the meaning of amer1 knockout groups is unclear. Is this F0 fish or F2 fish injected with upf3a MO. This information needs to be added to all figures.  

·      While describing quantitative changes in dlx2a (line 216, legend figure 7), the authors use the word “significant”/”significantly” what is incorrect since they did not performed statistical analysis.

·      Figure 8 and 10: b-catenin immunostaining data are not convincing. 1) It is unclear the plane of section; 2) the area being analyzed; 3) sample size; 4) images too small to interpret.

·      All whole mount images require either a higher mag image or insets of the craniofacial region to show defects, changes in gene expression etc (Figure 2, 3, 5, 7, 8, 9).

·      The authors state: “all experiments were performed in triplicate” (line 433). Because of variability when analyzing F0 fish, rescue assays, injection of MO, this sample size is insufficient.

·      The authors state that knockout efficacy was validated by PCR (data not shown) (line 131). This data needs to be included either in the manuscript or supplemental data because it is surprising that CRISPR on F0 fish resulted in total loss of amer1 in all analyzed fish. Most of the time, F0 crispants show knock down of the target gene.  

Reviewer 2 Report

Comments and Suggestions for Authors

This manuscript examines the function of the Amer1 gene in zebrafish craniofacial development and provides some evidence that effects of Amer1 knockout can be explained by increased Wnt signaling. The topic of this ms. is relevant to both understanding the etiology of microtia-atresia as well as basic developmental mechanisms involving Wnt signaling and should be of general interest to IJMS readers. However, there are a number of problems with this work as presented:

1) Lack of phenotype in F1 Amer1-/-. Authors state the initial F0 mutants obtained after CRISPR/Cas9 knockout are mosaic. Breeding of -/- offspring  from these did not have the craniofacial phenotype of F0-authors attribute this to genetic compensation and use a Upf3a morpholino to overcome this and then see the pharangeal arch phenotype seen in the F0 mutants. Since other Amer genes were not reported to suppress Wnt signaling like Amer1, it is hard to imaging how compensation would take place. Also, if compensation is taking place, why do F0 mosaic mutants have a phenotype. The authors should consider an alternative approach such as anti Amer1 morpholinos as a way to show that loss of Amer1 is the cause of the abnormal pharangeal arch development.

2) Since Amer1-/-/Upf3a morpholino mutants could not be bred, what was used in the remaining expts-F0 mosaic mutants or F1 Amer1-/- with morpholino treatment?  

3) Western blot studies in Fig 8A & 10A showing beta catenin elevation in Amer1-/- need to be quantified. Also in G, nuclear accumulation of beta catenin in Amer1-/- mutants is not clear. Red staining appears cytoplasmic in WT and mutant. Also, IF and subcellular distribution data should be quantified. Similarly, in Fig 10C, IWR-1-endo treated samples have more cytoplasmic beta catenin  staining, but nuclear accumulation is not apparent. It is also recommended that a more comprehensive analysis of changes in gene expression be included including more chondrocyte and bone markers.

4) In rescue experiments with IWR-1-endo, controls are only shown in whole mounts (Fig 9). Alcian blue images of pharangeal arches in 9D only show Amer1 knockouts with inhibitor. Need to show all conditions: WT +/- IWR-1-endo, mut +/- IWR-1-endo mut 

Minor

fig 6 BD-Y axes need labels

Round 2

Reviewer 1 Report

Comments and Suggestions for Authors

Although the authors addressed most of my comments, there are two major points that need to be addressed prior to publication:

- Calling a  F0 mosaic (line 175) crispant a knockout (related to question 9): I understand the authors provided plenty of evidence that all sgRNAs were generating deletions and insertions as shown in Fig. 3B-E and in additional information provided with the response. However, as they even stated in the paper, F0 zebrafish are mosaic (line 175) and not all cells will present the same type of deletion and/or insertion; some cells may not even have editing at all and remain "wild type". A "knockout" embryo, means the gene is not being expressed in all cells ("a global knockout"). The author stated that they checked mRNA levels by PCR (data not shown, line 131). A full global knockout would would mean that CRISPR editing generated insertions and deletions in all cells that caused changes in mRNA that led to non-sense mediated decay. However, it is hard to believe that different levels of editing (insertion and deletion; please check Fig. 3D for target R3; at least 4-5 types of different editing) would lead to similar changes in mRNA in all cells. Changes are random. Thus why if the authors want to continue to call F0 mosaic zebrafish as knockout, they need to provide scientific evidence that mRNA or protein is 100% reduced by quantitative real time PCR or western blot. They cannot just add a sentence with "data not shown". F0 mosaic embryos are KNOCK DOWN embryos; since some cells will be knockout while others will not. This is basic genetics. The reason we will breed out F0 animals to F1 and F2 is to pass the editing that happened in germline cells; then screen F2 animals for the editing that leads to a "knockout". The correct terminology is "F0 mosaic knockdown zebrafish". The F2 line is likely a knockout line. 

-b-catenin immunostaining: while the additional information is helping understand the area that was stained, the images are still of poor quality and not suitable for publication. The authors state this was done by confocal microscopy but there is so much out of focus light that the pictures look like regular fluorescence microscopy. 

Reviewer 2 Report

Comments and Suggestions for Authors

I am satisfied authors have addressed all issues raised in the previous critique.

minor Fig B y axis label is misspelled. Should be "Proliferative"

Round 3

Reviewer 1 Report

Comments and Suggestions for Authors

The authors addressed all concerns. The article is now suitable for publication.
